# DGG-HMR: Multi-Person Human Mesh Recovery with Depth-Guided Geometric Anchoring

Yanjie Li [1]   Le Hui [2 3]   Yali Peng [1]   Shigang Liu [1 †]

## Abstract

Multi-person human mesh recovery (HMR) from a single image is inherently ill-posed, as multiple 3D poses can produce identical 2D projections due to depth ambiguity. Most existing methods implicitly regress 3D translation from image features, which often leads to unreliable depth estimation. To address this issue, we propose a depth-guided multi-person HMR framework that explicitly models instance-level depth cues and integrates them into mesh recovery. Specifically, we first introduce an instance-aware depth estimator to predict per-person pelvis depths that serve as explicit 3D anchors, thereby decoupling depth estimation from mesh regression. Then, we design a geometry-anchored refinement decoder that uses these anchors to initialize each instance within a plausible 3D neighborhood, stabilizing mesh refinement under joint 2D-3D supervision. Finally, we adopt a single-stage joint training strategy to coordinate depth estimation and mesh recovery in a unified framework. Extensive experiments on multiple benchmarks demonstrate that our method achieves state-of-the-art performance in both mesh reconstruction accuracy and depth ordering. Code and models are available at https://github.com/Nebulae411/DGG-HMR.

## 1. Introduction

Multi-person human mesh recovery (HMR) from a single image is a challenging task in 3D human understanding. The task aims to reconstruct accurate 3D human meshes and

[1]School of Artificial Intelligence and Computer Science, Shaanxi Normal University [2]School of Electronics and Information & Shaanxi Key Laboratory of Information Acquisition and Processing, Northwestern Polytechnical University [3]Department of Computing, The Hong Kong Polytechnic University, Hong Kong, China. Correspondence to: Shigang Liu <shgliu@snnu.edu.cn>.

*Proceedings of the 43rd International Conference on Machine Learning*, Seoul, South Korea. PMLR 306, 2026. Copyright 2026 by the author(s).

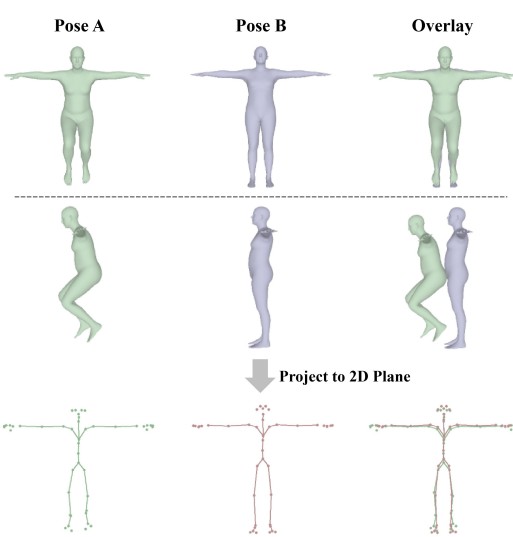

*Figure 1.* Illustration of the ill-posed problem in monocular HMR. Distinct 3D human configurations at different depths yield nearly identical 2D projections. From top to bottom, the figure shows the front view, side view, and 2D projection, respectively.

infer the spatial layout of all individuals in a scene (Sun et al., 2021; 2022; Baradel et al., 2024; Su et al., 2025). It is widely applied in fields such as AR/VR and interactive applications (Zhang et al., 2023b; Xu et al., 2023). In these scenarios, consistent spatial layouts and reliable relative depth ordering are particularly important. A dominant line of research follows a detect-and-crop pipeline, where each person is first detected and then reconstructed independently (Kanazawa et al., 2018; Kocabas et al., 2021; Goel et al., 2023; Zhang et al., 2021; 2023a; Li et al., 2022). Although these methods can improve single-person reconstruction accuracy, they lose global context in multi-person scenes, which often leads to inconsistent spatial layouts and incorrect depth ordering.

To overcome these limitations, one-stage methods (Sun et al., 2021; 2022; 2024; Baradel et al., 2024; Su et al., 2025) have recently emerged as a promising alternative. By encoding the full image, these methods preserve global context and support joint reasoning over multiple individuals within a unified image space. Moreover, benefiting from advances in vision transformers (ViTs) (Vaswani et al.,

2017; Dosovitskiy et al., 2021; Oquab et al., 2024) and the availability of large-scale, high-quality synthetic datasets (Patel et al., 2021; Black et al., 2023), one-stage methods have achieved substantial improvements in reconstruction accuracy. In particular, inspired by the success of DETR (Carion et al., 2020; Liu et al., 2022) in object detection, several works (Sun et al., 2024; Baradel et al., 2024; Su et al., 2025) introduce a DETR-style decoder into multi-person HMR. By designing instance-level queries and decoders, these methods achieve state-of-the-art performance in mesh reconstruction within a unified framework.

However, in monocular vision, depth ambiguity gives rise to an inherently ill-posed optimization problem. As shown in Figure 1, substantially different 3D human poses at varying depths may result in almost identical 2D projections. This depth uncertainty often drives the model toward sub-optimal solutions during optimization. Some prior works mitigate this ambiguity using additional depth or scene information, such as RGB-D sensor depth (Bashirov et al., 2021) and 3D scene geometry (Shen et al., 2023), while camera-aware monocular methods such as Zolly (Wang et al., 2023) and BLADE (Wang et al., 2025) study the effects of person depth, focal length, and perspective projection on mesh recovery. Nevertheless, these methods either require additional depth or scene input, or mainly focus on camera-aware single-person reconstruction, leaving instance-level depth constraints underexplored for crowded multi-person scenes. Most existing one-stage multi-person HMR methods (Baradel et al., 2024; Su et al., 2025) estimate depth, pose, and shape jointly from shared image features, without explicitly constraining the solution space. This makes inferring instance-level depth from a single full image extremely difficult. Moreover, depth uncertainty makes the joint optimization of 2D reprojection and 3D pose losses highly non-convex. The model may place a human at an incorrect depth and compensate by distorting pose or scale, while still minimizing both losses. As a result, the model struggles to anchor each individual to a correct 3D neighborhood, leading to unstable depth ordering and spatial layout.

In this work, we propose Multi-Person Human Mesh Recovery with Depth-Guided Geometric Anchoring (DGG-HMR), a framework that explicitly models instance-level depth cues and incorporates them into a geometry-anchored mesh refinement process. Our key observation is that explicitly constraining the 3D optimization starting point of each individual before pose decoding can effectively reduce depth uncertainty. Specifically, we first introduce an instance-aware depth estimator, which directly predicts instance-level pelvis depths from the full image. In addition, the depth estimator also outputs instance-level bounding boxes, which, together with the pelvis depths, form the geometric anchors used in our method. Building upon these geometric anchors,

we then design a geometry-anchored refinement decoder. This decoder provides an instance-aware 3D initialization, anchoring the optimization process in a locally consistent 3D neighborhood and alleviating depth ambiguity. Finally, we adopt a single-stage end-to-end training strategy to coordinate mesh recovery and depth estimation. Extensive experiments on 3DPW, CMU Panoptic, and MuPoTS-3D demonstrate that our method significantly improves depth ordering accuracy, achieving state-of-the-art reconstruction performance across all benchmarks.

The main contributions of this work are summarized as follows:

- We propose a depth-guided multi-person HMR framework that explicitly incorporates per-person depth to alleviate depth ambiguity.

- We introduce an instance-aware depth estimator that estimates instance-level depth, and use these depths to construct a geometry-anchored refinement decoder that initializes mesh optimization within instance-specific 3D neighborhoods, thereby stabilizing 2D-3D supervision under depth ambiguity.

- Extensive experiments demonstrate that the proposed DGG-HMR consistently outperforms state-of-the-art methods in mesh reconstruction accuracy and depth ordering.

## 2. Related Work

### 2.1. Multi-stage Human Mesh Recovery

Existing multi-person HMR methods can be broadly categorized into multi-stage and one-stage paradigms. Most multi-stage methods (Kanazawa et al., 2018; Goel et al., 2023; Zhang et al., 2021; 2023a) first employ mature detectors (Redmon & Farhadi, 2018; Ren et al., 2015; Carion et al., 2020) to obtain person bounding boxes. They then crop human regions from the full image and resize them to a unified resolution for regression. Such methods can preserve high-resolution human details and achieve strong accuracy in single-person scenes. However, their computational cost increases rapidly in crowded scenes. Occlusion handling and depth ordering also remain challenging.

### 2.2. One-stage Human Mesh Recovery

To overcome the limitations of multi-stage pipelines, recent studies have increasingly shifted toward one-stage methods (Sun et al., 2021; 2022; 2024; Baradel et al., 2024; Su et al., 2025). These methods regress all human meshes directly from the full image, avoiding the loss of global context caused by explicit cropping. ROMP (Sun et al., 2021) first proposed to recover multi-person meshes from a full im-

age. It localizes human centers via a body-center heatmap and regresses meshes from the corresponding indexed features. Subsequently, BEV (Sun et al., 2022) extends the body-center heatmap to a bird's-eye-view, enabling explicit reasoning about 3D relative positions within the scene. Recently, several works have introduced DETR-style pipelines into multi-person HMR, including Multi-HMR (Baradel et al., 2024), AiOS (Sun et al., 2024), and SAT-HMR (Su et al., 2025). These methods utilize learnable queries to represent human instances and decode mesh parameters from global image features. Benefiting from large-scale synthetic datasets (Patel et al., 2021; Black et al., 2023) and powerful visual backbones (Caron et al., 2021; Oquab et al., 2024), DETR-style methods have achieved state-of-the-art reconstruction accuracy in multi-person scenes.

Despite these advances, the ill-posed nature caused by depth ambiguity remains unresolved. TokenHMR (Dwivedi et al., 2024) improves HMR optimization by introducing the Threshold-Adaptive Loss Scaling (TALS) function. However, this method optimizes at the loss level and does not explicitly constrain depth uncertainty during projection. BLADE (Wang et al., 2025) further shows that depth estimation errors dominate projection inaccuracies, while focal length plays a comparatively minor role. These findings indicate that accurate instance-level depth estimation is essential for reducing ambiguity and stabilizing joint 2D-3D optimization in monocular multi-person HMR.

## 2.3. Monocular Depth Estimation

Monocular depth estimation aims to infer dense scene geometry from a single RGB image. Early monocular depth methods explored supervised learning from RGB-depth pairs and self-supervised learning from stereo or monocular videos (Eigen et al., 2014; Godard et al., 2017; 2019). Subsequent works improved robustness and cross-dataset generalization through stronger architectures and mixed-dataset training (Ranftl et al., 2022). More recent studies focus on metric-consistent depth estimation by modeling camera parameters and absolute scale (Yin et al., 2023; Piccinelli et al., 2024). In parallel, monocular depth estimation has shifted toward large-scale training and foundation-model-based paradigms. ZoeDepth (Bhat et al., 2023) unifies relative and metric depth prediction, improving cross-dataset generalization. Depth Anything (Yang et al., 2024a) shows that robust depth representations can be learned from massive unlabeled data. Building on this, Depth Anything V2 (Yang et al., 2024b) enhances structural coherence and spatial consistency via large-scale mixed-data training and distillation. Depth Pro (Bochkovskii et al., 2025) advances metric depth estimation with stronger scale consistency using modern backbones. These advances make monocular depth estimation reliable enough to serve as explicit geometric priors for high-level 3D reasoning tasks.

## 3. Preliminary

### 3.1. Body Model

We adopt SMPL (Loper et al., 2015) as our body model $\mathcal{M}$, which represents a human body using pose parameters $\boldsymbol{\theta} \in \mathbb{R}^{24 \times 3}$ and shape parameters $\boldsymbol{\beta} \in \mathbb{R}^{10}$. Specifically, the pose parameters $\boldsymbol{\theta}$ encode the relative rotations of 24 body joints in axis-angle form, and the shape parameters $\boldsymbol{\beta}$ define body proportions by combining learned shape bases. Given $\boldsymbol{\theta}$ and $\boldsymbol{\beta}$, SMPL outputs a triangulated human mesh $\mathbf{M} = \mathcal{M}(\boldsymbol{\theta}, \boldsymbol{\beta}) \in \mathbb{R}^{6890 \times 3}$. The 3D joints $\mathbf{X} \in \mathbb{R}^{J \times 3}$ are obtained via a linear joint regressor $\mathbf{X} = \mathbf{WM}$, where $\mathbf{W} \in \mathbb{R}^{J \times 6890}$ is a joint regression matrix and $J$ is the number of joints.

### 3.2. Camera Model and Perspective Projection

In this work, we adopt a standard pinhole camera model to relate 3D human geometry to the 2D image plane. Following common practice in HMR (Kanazawa et al., 2018; Goel et al., 2023), the predicted human mesh is represented in a root-relative coordinate system, where the root joint is defined as the pelvis. To recover the absolute position of each person in the camera coordinate system, we explicitly estimate a global translation vector that maps the root-relative mesh to the camera coordinate system. Let $\mathbf{J}_i^{\text{local}} = (x_i, y_i, z_i) \in \mathbb{R}^3$ denote the $i$-th 3D joint coordinate relative to the root joint. The global translation in the camera coordinate is defined as $\mathbf{T} = (T_x, T_y, T_z)$, where the depth component $T_z$ corresponds to the distance between the pelvis and the camera plane, and thus represents the overall depth of the human body. The absolute 3D position $\mathbf{J}_i^{\text{cam}}$ in the camera coordinate system is obtained by applying this translation:

$$\mathbf{J}_i^{\text{cam}} = \mathbf{J}_i^{\text{local}} + \mathbf{T} = (x_i + T_x, y_i + T_y, z_i + T_z). \quad (1)$$

Moreover, we define the camera focal length $f$ using a preset field of view (FoV) and the image size. Let $S_{hr}$ be the length of the longer image side and we set FoV $= 60°$. The focal length is calculated as $f = S_{hr}/(2 \tan(\text{FoV}/2))$. Assuming the principal point $(p_u, p_v)$ is located at the image center, the perspective projection $\Pi : \mathbb{R}^3 \to \mathbb{R}^2$ maps a 3D joint $(X, Y, Z) = \mathbf{J}_i^{\text{cam}}$ to its 2D pixel coordinates $(u, v)$:

$$\begin{aligned} u &= f \cdot \frac{X}{Z} + p_u = f \cdot \frac{x_i + T_x}{z_i + T_z} + p_u, \\ v &= f \cdot \frac{Y}{Z} + p_v = f \cdot \frac{y_i + T_y}{z_i + T_z} + p_v. \end{aligned} \quad (2)$$

### 3.3. Depth Ambiguity and Optimization Conflicts

We analyze two cases that reveal the fundamental conflict between 2D alignment and 3D pose accuracy.

**Case 1: Depth-Scale Ambiguity.** The first case arises from the coupled prediction of root-relative body geome-

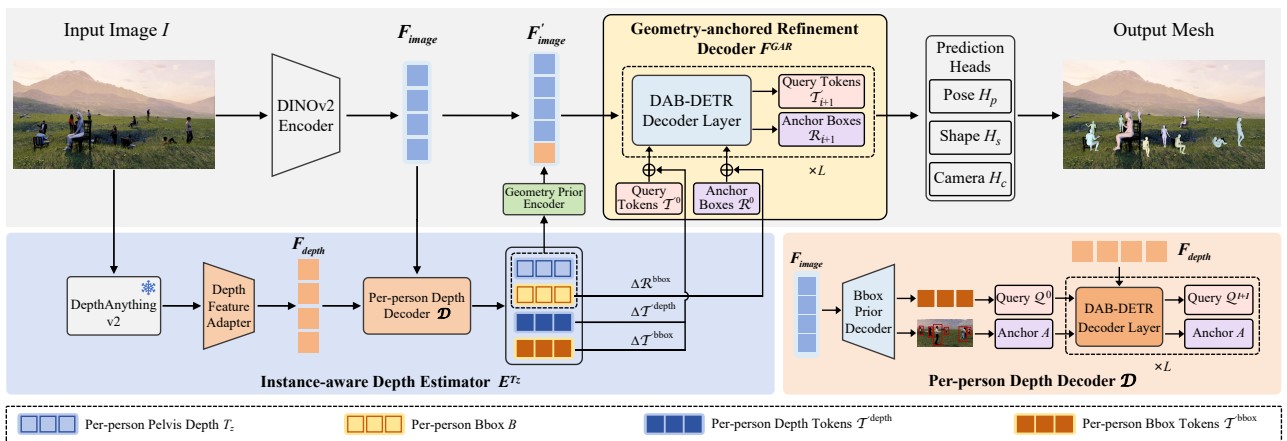

*Figure 2.* Overview of the proposed DGG-HMR framework. DGG-HMR adopts a dual-stream architecture that decouples instance-aware depth estimation and human mesh recovery. Image features $F_{\text{image}}$ are first extracted by a DINOv2 (Oquab et al., 2024) encoder and used to infer instance-level bounding boxes $B$ and corresponding tokens $\mathcal{T}^{\text{bbox}}$. In the depth stream, depth-aware features $F_{\text{depth}}$ extracted by Depth Anything V2 (Yang et al., 2024b) are combined with the instance anchors $B$ to estimate per-person pelvis depths $T_z$ and depth tokens $\mathcal{T}^{\text{depth}}$ via a DETR-style depth decoder $\mathcal{D}$. Based on the estimated geometric priors $(B, T_z)$, the HMR stream applies a geometry-anchored refinement decoder that injects these priors into decoder queries and reference points at initialization, and regresses SMPL pose, shape, and camera parameters to obtain the final meshes.

try and global translation. Since root-relative joints and global translation are predicted jointly, the model can scale them together during optimization. Let $\alpha$ denote such a coupled scaling factor. If the predicted local joints and global translation satisfy $\mathbf{J}_i^{\text{local}\prime} = \alpha \mathbf{J}_i^{\text{local}}$ and $\mathbf{T}' = \alpha \mathbf{T}$, then the resulting 2D projection remains unchanged:

$$u' = f \cdot \frac{\alpha x_i + \alpha T_x}{\alpha z_i + \alpha T_z} + p_u = f \cdot \frac{x_i + T_x}{z_i + T_z} + p_u = u. \quad (3)$$

The same argument applies to $v$. In this case, the reprojection error can remain nearly unchanged even though the recovered 3D body scale and absolute position are incorrect. This ambiguity indicates that, without explicit depth constraints, 2D supervision alone cannot uniquely determine the scale and depth of each person.

**Case 2: Pose Distortion under Incorrect Depth.** A more detrimental case occurs when the depth is estimated incorrectly while the global scale is implicitly constrained (e.g., by shape or bone-length priors). Assume the ground-truth depth is $Z_{\text{gt}}$, but the model predicts $Z_{\text{pred}} \neq Z_{\text{gt}}$. To satisfy the reprojection constraint $u_{\text{pred}} \approx u_{\text{gt}}$, the lateral coordinates must be distorted:

$$f \cdot \frac{X_{\text{pred}}}{Z_{\text{pred}}} \approx f \cdot \frac{X_{\text{gt}}}{Z_{\text{gt}}} \Rightarrow X_{\text{pred}} \approx X_{\text{gt}} \cdot \frac{Z_{\text{pred}}}{Z_{\text{gt}}}. \quad (4)$$

Here, the ratio $\gamma = \frac{Z_{\text{pred}}}{Z_{\text{gt}}}$ acts as a distortion factor. When the depth is overestimated ($\gamma > 1$), the model must enlarge the lateral joint offsets to match the 2D evidence. When the depth is underestimated ($\gamma < 1$), the 3D pose becomes compressed. Therefore, minimizing the 2D reprojection error under an incorrect depth hypothesis inevitably induces systematic pose distortion.

These two cases demonstrate that the joint optimization of 2D reprojection and 3D pose under ambiguous depth is an ill-posed and highly non-convex problem. Therefore, reliable instance-level depth estimation is essential for stabilizing 3D human mesh recovery.

## 4. Method

### 4.1. Overview

Given a single RGB image, our goal is to recover accurate 3D human meshes by reducing the depth ambiguity inherent in monocular reconstruction. Therefore, we propose a depth-guided multi-person HMR method that explicitly incorporates per-person depth cues into the reconstruction process. The overview of the proposed method (DGG-HMR) is illustrated in Figure 2. DGG-HMR adopts a dual-stream architecture that decouples human mesh regression and instance-level depth estimation. We first propose an instance-aware depth estimator that directly regresses instance-level pelvis depths from the entire image (Sec. 4.2). Specifically, this estimator leverages bounding boxes inferred from DINOv2 image features as fixed human anchors and predicts instance-level pelvis depth by integrating human anchors with depth-aware features. Based on these geometric priors, we further design a geometry-anchored refinement decoder to refine human meshes within a physically plausible spatial domain (Sec. 4.3). Finally, we introduce a single-stage joint training strategy to balance depth estimation and human mesh recovery under a shared backbone (Sec. 4.4), and define the corresponding training objectives and loss functions in Sec. 4.5.

## 4.2. Instance-aware Depth Estimation

We introduce an instance-aware depth estimator $E^{T_z}$ to predict per-person pelvis depth $T_z$. Given an RGB image $I$, we first extract global depth-aware features $F_{\text{depth}} = \phi(I) \in \mathbb{R}^{H' \times W' \times C}$ using Depth Anything V2 (Yang et al., 2024b). Since Depth Anything V2 predicts relative depth, we further apply a depth feature adapter (a few ConvNet layers) to transform $F_{\text{depth}}$ into features suitable for anchor-conditioned depth prediction. In parallel, we infer instance-level human bounding boxes $B = \{B_i\}_{i=1}^N$ from DINOv2 image features $F_{\text{image}}$ using a lightweight DETR-style bounding box decoder. This decoder also outputs a set of bounding box tokens, denoted as $\mathcal{T}^{\text{bbox}} = \{\mathbf{t}_i^{\text{bbox}}\}_{i=1}^N$. Each bounding box $B_i$ is treated as a fixed anchor $A_i$, which serves as a spatial reference for depth prediction. We denote the anchor set as $A = \{A_i\}_{i=1}^N$. To estimate per-person pelvis depth, we design a DETR-style per-person depth decoder $\mathcal{D}$. Specifically, the bounding box tokens $\mathcal{T}^{\text{bbox}}$ are used to initialize the depth queries $\mathcal{Q}^0 = \mathcal{T}^{\text{bbox}}$, and the queries at the $l$-th decoding layer are denoted as $\mathcal{Q}^l = \{\mathbf{q}_i^l\}_{i=1}^N$. The depth decoder $\mathcal{D}$ iteratively refines the queries through cross-attention with the depth-aware feature $F_{\text{depth}}$ under the guidance of the fixed anchors:

$$\mathcal{Q}^{l+1} = \mathcal{D}(\mathcal{Q}^l, F_{\text{depth}}, A), \quad l = 0, \ldots, L-1. \quad (5)$$

The final pelvis depth is predicted from the output queries at the last decoding layer:

$$T_z = E^{T_z}(A, \mathcal{Q}^L, F_{\text{depth}}), \quad (6)$$

where $\mathcal{Q}^L$ denotes the refined depth query set at the final decoding layer, and $T_z$ represents the predicted depths (defined as pelvis depth) for all persons in the image. By performing anchor-conditioned prediction, $E^{T_z}$ can directly infer per-person depth from full-image features without relying on external detectors or cropped inputs.

For depth supervision, we follow the design in BLADE (Wang et al., 2025). We weight the depth error inversely proportional to the ground-truth depth, resulting in a weighted $L1$ loss:

$$\mathcal{L}_{\text{depth}} = \frac{1}{T_z^{\text{GT}}} \left| T_z - T_z^{\text{GT}} \right|. \quad (7)$$

This weighting strategy encourages the model to focus on accurate depth estimation for closer subjects, which are more sensitive to depth errors.

## 4.3. Geometry-anchored Refinement Decoder

In Section 4.2, we estimate per-person depths $T_z$ using an instance-aware depth estimator and simultaneously obtain human bounding boxes $B$. Following common practice, we encode the geometric priors of all detected instances into

a scene-level geometry representation. Specifically, the set of bounding boxes $B = \{B_i\}_{i=1}^N$ and their corresponding pelvis depths $T_z = \{T_{z,i}\}_{i=1}^N$ are jointly embedded by a lightweight geometry encoder. The resulting geometry representation is concatenated with DINOv2 image tokens to construct geometry-aware image features:

$$F'_{\text{image}} = \text{Concat}\left(F_{\text{image}}, \mathcal{E}(B, T_z)\right), \quad (8)$$

where $\mathcal{E}(\cdot)$ aggregates scene-level box-depth priors into the feature space.

While this enriches image features with geometric cues, we find that such feature-level fusion provides limited benefit for resolving depth ambiguity, since the injected geometry is easily diluted among dense image tokens and does not directly influence the mesh regression. To address this, we propose a DETR-style geometry-anchored refinement decoder $F^{\text{GAR}}$, which injects instance-level geometric priors directly into the decoder queries. Specifically, the depth estimator $E^{T_z}$ outputs the set of human bounding boxes $B$, the set of bounding box tokens $\mathcal{T}^{\text{bbox}}$, and the set of depth-aware tokens $\mathcal{T}^{\text{depth}}$, collectively representing all instances in the image. Let $\mathcal{T}^0$ and $\mathcal{R}^0$ denote the sets of initial query tokens and reference points in the refinement decoder. Instance-level geometric priors are injected into these sets via zero-initialized residual additions:

$$\mathcal{T}^0 = \mathcal{T}_{\text{base}}^0 + \Delta \mathcal{T}^{\text{bbox}} + \Delta \mathcal{T}^{\text{depth}}, \quad (9)$$

$$\mathcal{R}^0 = \mathcal{R}_{\text{base}}^0 + \Delta \mathcal{R}^{\text{bbox}}, \quad (10)$$

where $\Delta \mathcal{T}^{\text{bbox}}$ and $\Delta \mathcal{T}^{\text{depth}}$ are the residual embeddings derived from the sets $\mathcal{T}^{\text{bbox}}$ and $\mathcal{T}^{\text{depth}}$, respectively, and $\Delta \mathcal{R}^{\text{bbox}}$ is computed from the bounding box set $B$. All residual terms are added only at the decoder input, allowing geometric priors to guide the optimization starting point for all instances without imposing hard constraints during iterative refinement.

## 4.4. Joint Optimization Training Strategy

Although we decouple instance-level depth estimation from mesh recovery, the depth branch still relies on the shared DINOv2 image features $F_{\text{image}}$ to obtain instance-level bounding boxes. Under the conventional two-stage training paradigm, the depth branch is first pre-trained to obtain a strong depth prior and then jointly fine-tuned with the mesh recovery branch. However, we observe that this strategy biases the shared backbone toward detection and localization, and degrades the feature quality for pose and shape regression. Therefore, we adopt a single-stage end-to-end training strategy to jointly optimize both branches from scratch. Specifically, in the depth branch, we freeze the Depth Anything V2 (Yang et al., 2024b) backbone and only train the depth feature adapter and the per-person depth decoder. Moreover, we explicitly stop the gradient flow from

*Table 1.* Comparison with SOTA methods on 3DPW (Von Marcard et al., 2018), CMU Panoptic (Joo et al., 2015), and MuPoTS-3D (Mehta et al., 2018) datasets. "Res." denotes the input image resolution. "1288*" indicates that SAT-HMR (Su et al., 2025) adopts a variable-resolution strategy with 1288 as the base resolution and switches to 644 when necessary. "Crop" means the method uses a crop-based input. Our method achieves superior overall performance.

| Method | Res. | 3DPW | | | CMU Panoptic (MPJPE) | | | | | MuPoTS | |
| --- | --- | --- | --- | --- | --- | --- | --- | --- | --- | --- | --- |
| | | PA-MPJPE↓ | MPJPE↓ | MVE↓ | Haggling↓ | Mafia↓ | Ultimatum↓ | Pizza↓ | Avg.↓ | PCK-All↑ | PCK-Matched↑ |
| CRMH (Jiang et al., 2020) | 832 | - | - | - | 129.6 | 133.5 | 153.0 | 156.7 | 143.2 | 69.1 | 72.2 |
| ROMP (Sun et al., 2021) | 512 | 47.3 | 76.6 | 93.4 | 110.8 | 122.8 | 141.6 | 137.6 | 128.2 | 69.9 | 72.2 |
| 3DCrowdNet (Choi et al., 2022) | Crop | 51.5 | - | 98.3 | 109.6 | 135.9 | 129.8 | 135.6 | 127.6 | 72.7 | 73.3 |
| BEV (Sun et al., 2022) | 512 | 46.9 | 78.5 | 92.3 | 90.7 | 103.7 | 113.1 | 125.2 | 109.5 | 70.2 | 75.2 |
| PSVT (Qiu et al., 2023) | 512 | 45.7 | - | 84.9 | 88.7 | 97.9 | 115.2 | 121.2 | 105.7 | - | - |
| Multi-HMR (Baradel et al., 2024) | 896 | 46.7 | 70.9 | 86.9 | - | - | - | - | 94.6 | 79.4 | 84.6 |
| SAT-HMR (Su et al., 2025) | 1288* | 41.6 | 63.6 | _73.7_ | **67.5** | 78.5 | 95.8 | 94.6 | 84.2 | **89.0** | **90.1** |
| DGG-HMR (ours) | 672 | _40.5_ | _62.9_ | 73.8 | 71.6 | _77.7_ | _83.6_ | _89.0_ | _80.8_ | 83.6 | _89.2_ |
| DGG-HMR (ours) | 896 | **39.0** | **61.1** | **70.9** | _69.3_ | **77.5** | **83.4** | **85.6** | **79.2** | _86.2_ | **90.1** |

the mesh recovery branch to the depth prediction head, so that the non-convex optimization of mesh recovery does not corrupt the depth estimation. With this design, the depth branch provides stable and reliable $T_z$ estimates as geometric anchors, which in turn stabilize the subsequent geometry-anchored mesh refinement.

*Table 2.* Comparison with SOTA methods on AGORA dataset. Note: Since the AGORA test server is unavailable, we conduct evaluations on the validation set with a nearly identical data distribution to the test set.

| Method | Res. | AGORA | | | | |
| --- | --- | --- | --- | --- | --- | --- |
| | | F1-score↑ | Precision↑ | Recall↑ | NMJE↓ | NMVE↓ |
| ROMP (Sun et al., 2021) | 512 | 0.91 | 0.95 | 0.88 | 118.8 | 113.6 |
| BEV (Sun et al., 2022) | 512 | 0.93 | 0.96 | 0.90 | 113.2 | 108.3 |
| PSVT (Qiu et al., 2023) | 512 | 0.93 | - | - | 105.1 | 101.2 |
| AiOS (Sun et al., 2024) | 1333 | _0.94_ | _0.98_ | 0.90 | **68.0** | **61.2** |
| Multi-HMR (Baradel et al., 2024) | 896 | 0.93 | - | - | 89.0 | 83.4 |
| SAT-HMR (Su et al., 2025) | 1288* | **0.95** | _0.98_ | _0.91_ | _71.5_ | _66.6_ |
| DGG-HMR (ours) | 672 | 0.93 | **0.99** | 0.87 | 80.5 | 75.7 |
| DGG-HMR (ours) | 896 | **0.95** | **0.99** | **0.92** | 72.6 | 68.4 |

### 4.5. Training Losses

Similar to DETR (Carion et al., 2020), we first establish a correspondence between predicted human queries and ground-truth instances using the Hungarian algorithm (Kuhn, 1955). The matching cost considers bounding box $L1$ and GIoU (Rezatofighi et al., 2019) errors, confidence scores, and projected 2D joint errors. The overall training loss combines contributions from the HMR and the depth branch. Each component is weighted to balance multi-task supervision under the shared backbone. The overall training loss $\mathcal{L}$ is the sum:

$$\mathcal{L} = \mathcal{L}_{\text{HMR}} + \mathcal{L}_{\text{depth-branch}}. \quad (11)$$

The HMR loss $\mathcal{L}_{\text{HMR}}$ supervises the prediction of SMPL parameters, 3D joints, 2D projections, and human queries:

$$\mathcal{L}_{\text{HMR}} = \lambda_{\text{pose}}\mathcal{L}_{\text{pose}} + \lambda_{\text{shape}}\mathcal{L}_{\text{shape}} + \lambda_{j3d}\mathcal{L}_{j3d} + \lambda_{j2d}\mathcal{L}_{j2d}$$
$$+ \lambda_{\text{box}}\mathcal{L}_{\text{box}} + \lambda_{\text{giou}}\mathcal{L}_{\text{giou}} + \lambda_{\text{conf}}\mathcal{L}_{\text{conf}} + \lambda_{\text{Tz}}\mathcal{L}_{\text{Tz}}, \quad (12)$$

where $\mathcal{L}_{\text{pose}}$ and $\mathcal{L}_{\text{shape}}$ are $L1$ losses on SMPL pose and shape parameters. $\mathcal{L}_{j3d}$ and $\mathcal{L}_{j2d}$ are $L1$ losses on 3D joints and their projected 2D locations. $\mathcal{L}_{\text{box}}$ and $\mathcal{L}_{\text{giou}}$ supervise the predicted human bounding boxes with $L1$ and GIoU losses (Rezatofighi et al., 2019), respectively. $\mathcal{L}_{\text{conf}}$ supervises the confidence scores of human queries using focal loss (Lin et al., 2017). Since the HMR branch performs projection with a preset focal length $f^{\text{pre}}$, its predicted depth is represented in the preset-camera coordinate scale rather than directly in the ground-truth camera scale. Therefore, we supervise it by converting the predicted depth back to the ground-truth focal-length scale (Facil et al., 2019):

$$\mathcal{L}_{T_z} = \frac{1}{T_z^{\text{GT}}} \left| \frac{T_z f^{\text{GT}}}{f^{\text{pre}}} - T_z^{\text{GT}} \right|. \quad (13)$$

For the depth branch, the loss combines depth supervision and the bbox-prior decoder losses:

$$\mathcal{L}_{\text{depth-branch}} = \lambda_{\text{depth}}\mathcal{L}_{\text{depth}} + \lambda_{\text{box}}\mathcal{L}_{\text{box}}$$
$$+ \lambda_{\text{giou}}\mathcal{L}_{\text{giou}} + \lambda_{\text{conf}}\mathcal{L}_{\text{conf}}. \quad (14)$$

Here, $\mathcal{L}_{\text{depth}}$ directly measures the per-person pelvis depth error in meters, without focal-length normalization. $\mathcal{L}_{\text{box}}$, $\mathcal{L}_{\text{giou}}$, and $\mathcal{L}_{\text{conf}}$ follow the same definitions as in the HMR branch.

## 5. Experiments

### 5.1. Datasets and Metrics

We train our model on a mixture of synthetic and real-world datasets, including AGORA (Patel et al., 2021), BEDLAM (Black et al., 2023), COCO (Lin et al., 2014), CrowdPose (Li et al., 2019), and MPII (Andriluka et al., 2014). For evaluation, we perform comprehensive experiments on multiple benchmarks and report metrics such as MPJPE, PA-MPJPE, MVE, NMJE, NMVE, PCDR, ADE, and PCK. Further details on dataset splits, benchmark settings, and metric definitions are provided in Appendix A.1 and A.2. Additional training details are provided in Appendix A.3.

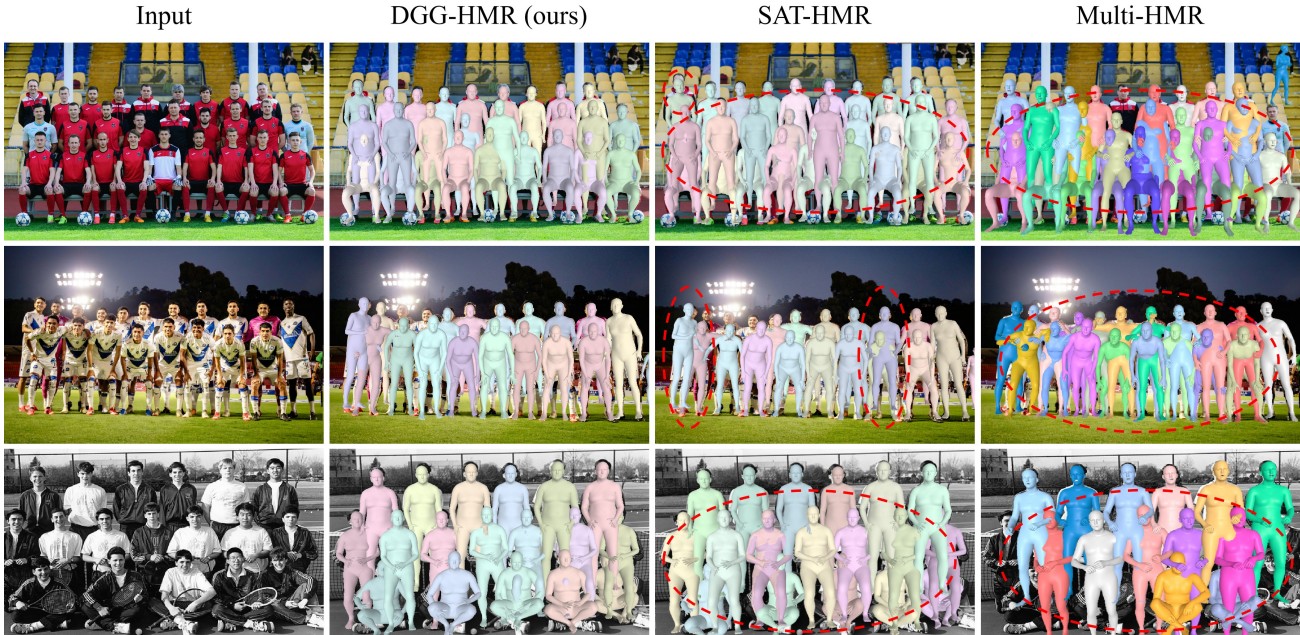

*Figure 3.* Comparison with SOTA methods on in-the-wild images from the Internet. This figure presents qualitative comparisons of our DGG-HMR with SAT-HMR (Su et al., 2025) and Multi-HMR (Baradel et al., 2024). Our method demonstrates more accurate spatial layout and relative depth ordering for multiple humans, and exhibits more robust performance in dense crowds and complex scenes.

*Table 3.* Depth ordering and absolute depth error on AGORA validation dataset. Our method achieves the best performance in both metrics across all depth intervals (0-2m, 2-5m, 5-10m, 10m+).

| Method | Res. | 0-2m | | 2-5m | | 5-10m | | 10m+ | |
|---|---|---|---|---|---|---|---|---|---|
| | | $PCDR^{0.2}(\%) \uparrow$ | ADE (m) $\downarrow$ | $PCDR^{0.2}(\%) \uparrow$ | ADE (m) $\downarrow$ | $PCDR^{0.2}(\%) \uparrow$ | ADE (m) $\downarrow$ | $PCDR^{0.2}(\%) \uparrow$ | ADE (m) $\downarrow$ |
| ROMP (Sun et al., 2021) | 512 | 95.31 | 1.981 | 89.32 | 3.014 | 87.98 | 3.400 | 80.33 | 3.101 |
| BEV (Sun et al., 2022) | 512 | 96.22 | 1.763 | 90.48 | 2.786 | 90.19 | 2.786 | 84.75 | 3.034 |
| Multi-HMR (Baradel et al., 2024) | 896 | 100.0 | 1.562 | 94.61 | 2.306 | 93.71 | 2.456 | 93.12 | 2.211 |
| SAT-HMR (Su et al., 2025) | 1288* | 100.0 | 1.494 | 95.23 | 2.199 | 94.53 | 2.360 | 94.02 | 2.007 |
| DGG-HMR (ours) | 672 | 100.0 | 1.455 | 95.90 | 2.196 | 94.63 | 2.357 | 93.64 | 2.000 |
| DGG-HMR (ours) | 896 | 100.0 | 1.430 | 96.01 | 2.224 | 94.83 | 2.351 | 94.38 | 1.943 |

## 5.2. Comparison to State-of-the-art Methods

**Quantitative Results.** We evaluate our method on four widely used benchmarks, including 3DPW (Von Marcard et al., 2018), AGORA (Patel et al., 2021), CMU Panoptic (Joo et al., 2015), and MuPoTS-3D (Mehta et al., 2018), and compare it with recent state-of-the-art approaches. Quantitative results are summarized in Table 1 and Table 2, and our method achieves strong and consistent performance across all datasets. On real-world datasets such as 3DPW, CMU Panoptic, and MuPoTS-3D, our approach consistently reports lower reconstruction errors than existing methods. AGORA is a large-scale synthetic dataset that includes highly crowded scenes and a large number of small-scale human instances, which makes accurate reconstruction particularly sensitive to input resolution. Notably, although our method is not specifically optimized for high-resolution inputs, it still achieves competitive performance compared to recent methods. Beyond standard mesh reconstruction

metrics, we further evaluate percentage of correct depth relations (PCDR) and absolute depth errors (ADE) on AGORA, as reported in Table 3. Our method outperforms prior approaches on both metrics, demonstrating more reliable instance-level spatial reasoning in crowded scenes and further confirming the effectiveness of explicitly modeling per-person depth in multi-person human mesh recovery.

**Visualization Results.** We further conduct qualitative evaluations on multiple benchmarks and Internet images. As shown in Figure 3, we compare our method with SOTA methods in crowded multi-person scenes. Our method consistently produces more accurate reconstructions and relative depth ordering. In challenging cases with heavy occlusions and large-scale variations, competing methods often miss instances or produce incorrect depth layouts, while our method maintains stable and coherent predictions. In addition, we visualize the reconstructed meshes from a top-down view to illustrate the recovered 3D spatial layout. As shown

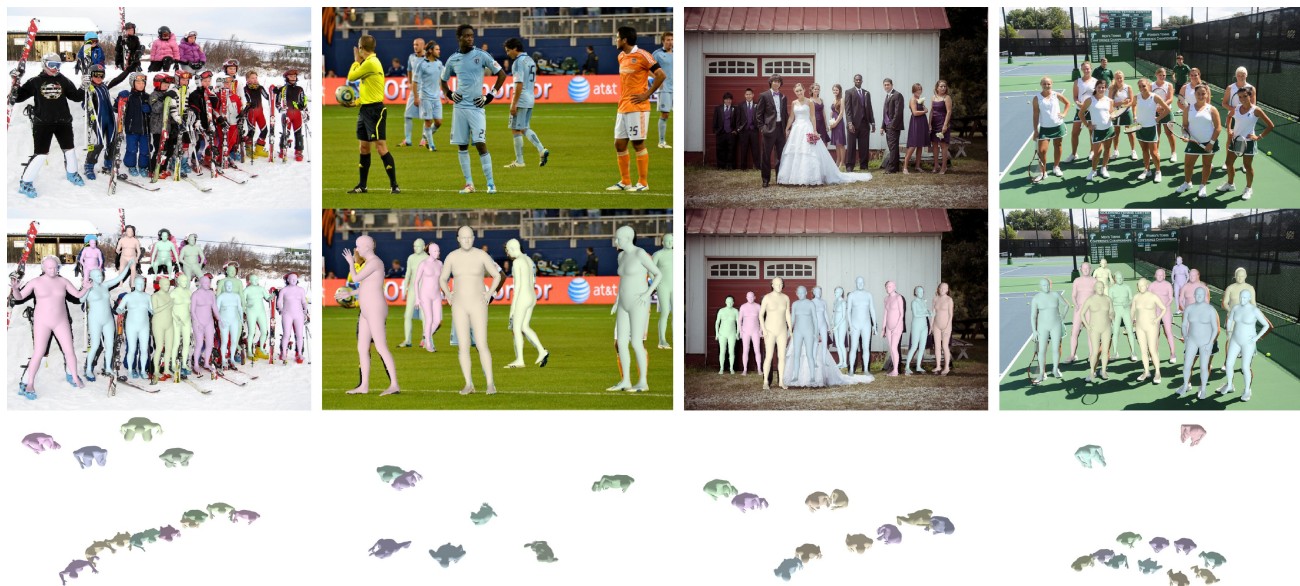

*Figure 4.* Qualitative results of our method on CrowdPose (Li et al., 2019) dataset. We visualize four dense crowd cases from the CrowdPose dataset, displaying the input image, estimated mesh overlay, and a top view of the 3D results (from top to bottom). The results demonstrate our method's robust performance in handling multi-person dense scenarios.

in Figure 4, our method produces a physically consistent ordering of multiple individuals in 3D space.

### 5.3. Ablation Study

We conduct ablation studies to analyze the contribution of the key components in our framework and the effectiveness of the proposed training strategy.

**Effect of Instance-aware Depth Estimator.** We first analyze the effects of the instance-aware depth estimator. Starting from a baseline without any explicit depth prior, we first introduce the instance-level depth by encoding the predicted $T_z$ as a token and concatenating it with image tokens. As shown in Table 4, this simple modification already brings clear improvements, verifying that instance-level depth priors effectively alleviate depth ambiguity. However, this naive depth token concatenation does not fully resolve the 2D-3D optimization conflict.

**Impact of Geometry-anchored Refinement Decoder.** When further enabling the geometry-anchored refinement decoder, we observe consistent gains across all metrics. This demonstrates that explicit geometric anchoring plays a critical role beyond simple feature augmentation, leading to more stable and spatially coherent reconstruction.

**Analysis of Training Strategies.** We compare different training strategies in Table 5. In the multi-stage strategy, we first train the instance-aware depth estimator. During this stage, the shared DINOv2 image encoder is optimized to predict human boxes and box tokens for depth anchoring. We then freeze the depth branch and train the mesh recovery

branch using the obtained depth anchors. Although this strategy produces reasonable depth estimates, the first stage tends to bias the shared image encoder toward detection and localization. Such features are less aligned with pose and shape regression, which limits the subsequent mesh recovery. In contrast, our single-stage joint training optimizes depth estimation and mesh recovery together from the beginning. This allows the shared image encoder to balance geometric localization and semantic reconstruction, leading to consistently better overall performance.

*Table 4.* Ablation study on the core components of our framework on the AGORA (Patel et al., 2021) dataset. IDE denotes the Instance-aware Depth Estimator, and GAR Decoder denotes the Geometry-Anchored Refinement Decoder.

| IDE ($T_z$ token) | GAR Decoder | MPJPE↓ | MVE↓ | PCDR$^{0.2}$(%)↑ | ADE↓ |
|:---:|:---:|:---:|:---:|:---:|:---:|
| ✗ | ✗ | 77.3 | 72.9 | 89.3 | 2.34 |
| ✓ | ✗ | 74.3 | 69.1 | 92.4 | 2.20 |
| ✓ | ✓ | **69.0** | **64.9** | **94.8** | **2.12** |

*Table 5.* Ablation study of different training strategies on the AGORA (Patel et al., 2021) dataset. Two-stage denotes the conventional strategy that first pre-trains the depth branch and then jointly fine-tunes the full network. Single-stage denotes our single-stage end-to-end joint training strategy.

| Training strategy | MPJPE↓ | MVE↓ | NMJE↓ | NMVE↓ | PCDR$^{0.2}$(%)↑ |
|:---|:---:|:---:|:---:|:---:|:---:|
| Two-stage | 75.3 | 71.1 | 79.7 | 75.9 | 93.9 |
| Single-stage (ours) | **69.0** | **64.9** | **72.6** | **68.3** | **94.8** |

# 6. Conclusion

In this work, we revisit monocular multi-person human mesh recovery from the perspective of depth ambiguity and its induced optimization conflicts. We reformulate the problem as a depth-guided, geometry-anchored refinement process. Specifically, we introduce an instance-aware depth estimator to predict per-person pelvis depth from the full image, and use these predictions as explicit geometric priors. Based on these priors, the subsequent mesh refinement is anchored within instance-specific 3D neighborhoods, which stabilizes the optimization under depth ambiguity. The entire framework is trained in a unified single-stage manner to coordinate depth estimation and mesh refinement. This is the first attempt to explicitly introduce instance-level depth priors into monocular multi-person HMR, providing a new perspective on resolving depth ambiguity and optimization conflict in multi-person scenes.

# Acknowledgements

This work is supported by the National Natural Science Foundation of China (No. 62477028, 62377032, 62306238), and the Fundamental Research Funds for the Central Universities.

# Impact Statement

This paper presents work whose goal is to advance the field of Machine Learning. There are many potential societal consequences of our work, none which we feel must be specifically highlighted here.

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

# A. Training and Evaluation Details

## A.1. Datasets

We train our model on a mixture of synthetic and real-world datasets, including AGORA (Patel et al., 2021), BEDLAM (Black et al., 2023), COCO (Lin et al., 2014), CrowdPose (Li et al., 2019), and MPII (Andriluka et al., 2014). AGORA and BEDLAM provide high-quality synthetic data with accurate 3D ground truth annotations, which support supervision for full 3D human mesh recovery. COCO, CrowdPose, and MPII contain only 2D annotations and are therefore used for 2D supervision.

**AGORA**. A photorealistic synthetic dataset featuring a wide spectrum of scenarios. Equipped with precise 3D human annotations derived from SMPL (Loper et al., 2015) and SMPL-X (Pavlakos et al., 2019), it has emerged as a standard benchmark for 3D human mesh recovery tasks. We utilize 14K training images (107K instances), 1K validation images (8K instances), and 3K test images from this dataset.

**BEDLAM**. A large-scale synthetic dataset characterized by rich variations in body shape, motion, skin tone, hairstyle, and clothing. Its original training split contains 286K images (951K instances), paired with 29K validation images. For our method, we downsample the training set to 1/6 of its original size, resulting in 48K images (159K instances). The test set is not used, as it currently lacks support for the SMPL format.

**COCO, CrowdPose, MPII**. These real-world multi-person datasets were initially designed for 2D human pose estimation. To enhance the generalization of our model to real scenarios, we extend them with 3D pseudo-annotations generated by NeuralAnnot (Moon et al., 2022). Following the data processing protocols of SAT-HMR (Su et al., 2025), we perform uniform downsampling on these datasets. Specifically, we retain 16K images (66K instances) for COCO, 10K images (36K instances) for CrowdPose, and 17K images (29K instances) for MPII.

**3DPW**. An in-the-wild dataset with 3D human mesh annotations. We fine-tune our model on its 17K training images, then perform final evaluation on 24K test images.

**MuPoTS-3D**. A real-world multi-person 3D pose dataset consisting of over 8K frames across 20+ scenes, each containing at least three subjects with annotated 3D poses. It is exclusively used to assess our model's generalization capability.

**CMU Panoptic**. An indoor multi-person dataset capturing 4 sequences of diverse group activities, with approximately 9K images. Like MuPoTS-3D, it is employed solely to validate the model's generalization performance.

## A.2. Evaluation Details

Following prior works (Baradel et al., 2024; Su et al., 2025), we evaluate our approach on multiple widely adopted benchmarks, including 3DPW (Von Marcard et al., 2018), AGORA (Patel et al., 2021), MuPoTS-3D (Mehta et al., 2018), and CMU Panoptic (Joo et al., 2015). These benchmarks cover diverse settings, ranging from outdoor scenes to indoor multi-person scenarios. On the 3DPW, reconstruction accuracy is evaluated using Mean Per-Joint Position Error (MPJPE), Procrustes-Aligned Mean Per-Joint Position Error (PA-MPJPE), and Mean Vertex Error (MVE). For AGORA, both detection and reconstruction quality are evaluated. Detection performance is measured using F1-score, Precision, and Recall, while reconstruction accuracy is assessed using Normalized Mean Joint Error (NMJE) and Normalized Mean Vertex Error (NMVE). To evaluate depth reasoning, we adopt the Percentage of Correct Depth Relations (PCDR$^{0.2}$) (Sun et al., 2022) with a threshold of 0.2 m and Absolute Depth Error (ADE). On MuPoTS-3D, we report Percentage of Correct Keypoints (PCK) for both matched individuals and all detected persons, following standard protocols (Sun et al., 2021; Baradel et al., 2024; Su et al., 2025). For CMU Panoptic, we report Mean Per-Joint Position Error (MPJPE) on four representative scenes and average the results across all evaluated sequences.

## A.3. Training Details

We train our models at a resolution of 672 on 6 NVIDIA L20 GPUs, taking approximately 3 days for the ViT-B backbone. We use the AdamW (Loshchilov & Hutter, 2017) optimizer with a weight decay of $1 \times 10^{-4}$. The initial learning rate is set to $4 \times 10^{-5}$, and a reduced rate of $2 \times 10^{-5}$ is applied to the backbone encoders. Training is conducted for 50 epochs using a cosine learning rate scheduler. Our total loss consists of several weighted components. We set the weights for confidence, body pose, and shape parameters to $\lambda_{\text{conf}} = 4.0$, $\lambda_{\text{pose}} = 5.0$, and $\lambda_{\text{shape}} = 3.0$, respectively. The 2D bounding box and GIoU losses use a weight of $\lambda_{\text{box}} = 2.0$ and $\lambda_{\text{giou}} = 2.0$. We assign weights of $\lambda_{j3d} = 8.0$ and $\lambda_{j2d} = 40.0$ to the 3D and 2D joint losses. In the depth branch, the depth loss ($\mathcal{L}_{\text{depth}}$) receives a weight of $\lambda_{\text{depth}} = 4.0$. The bbox-prior decoder losses for boxes, GIoU, and confidence are $\lambda_{\text{box}} = 0.5$, $\lambda_{\text{giou}} = 0.5$, and $\lambda_{\text{conf}} = 1.0$, respectively.

# B. Analysis of Depth Losses

## B.1. Discussion on Depth Loss Choices

In monocular human mesh recovery, depth errors affect 2D reprojection through the perspective projection:

$$u = f\frac{x}{z}, \quad v = f\frac{y}{z}. \tag{15}$$

The sensitivity of the projected coordinates with respect to depth is:

$$\frac{\partial u}{\partial z} = -f\frac{x}{z^2}, \quad \frac{\partial v}{\partial z} = -f\frac{y}{z^2}, \tag{16}$$

which indicates that depth errors at smaller $z$ lead to much larger reprojection deviations, while the same depth error at large $z$ has a significantly smaller effect on 2D alignment.

Let $T_z$ denote the predicted pelvis depth and $T_z^{\mathrm{GT}}$ the ground-truth depth. We consider three commonly used depth regression losses:

**Absolute depth loss.** This loss assigns equal weight to depth errors at all distances, although their impacts on 2D reprojection differ significantly.

$$\mathcal{L}_{\mathrm{abs}} = \left| T_z - T_z^{\mathrm{GT}} \right|. \tag{17}$$

**Inverse-depth loss.**

$$\mathcal{L}_{\mathrm{inv}} = \left| \frac{1}{T_z} - \frac{1}{T_z^{\mathrm{GT}}} \right|. \tag{18}$$

Using first-order Taylor approximation around $T_z^{\mathrm{GT}}$, we obtain:

$$\mathcal{L}_{\mathrm{inv}} \approx \frac{1}{(T_z^{\mathrm{GT}})^2} \left| T_z - T_z^{\mathrm{GT}} \right|. \tag{19}$$

Therefore, inverse-depth loss implicitly applies a weighting factor proportional to $1/(T_z^{\mathrm{GT}})^2$, which strongly emphasizes nearby subjects but may cause excessively large gradients when $T_z^{\mathrm{GT}}$ is small.

**Depth-weighted loss (ours).** We adopt a weighted $L1$ loss of the form:

$$\mathcal{L}_{\mathrm{w}} = \frac{1}{T_z^{\mathrm{GT}}} \left| T_z - T_z^{\mathrm{GT}} \right|. \tag{20}$$

This formulation can be regarded as a compromise between absolute depth regression and inverse-depth supervision. It still assigns larger weights to closer subjects, which are more sensitive in perspective projection, while avoiding the overly aggressive weighting of $1/(T_z^{\mathrm{GT}})^2$ used by inverse-depth loss.

The three losses can be interpreted as applying different depth-dependent weights to the absolute error:

$$\mathcal{L}_{\mathrm{abs}} \propto 1 \cdot |\Delta T_z|, \tag{21}$$

$$\mathcal{L}_{\mathrm{w}} \propto \frac{1}{T_z^{\mathrm{GT}}} |\Delta T_z|, \tag{22}$$

$$\mathcal{L}_{\mathrm{inv}} \propto \frac{1}{(T_z^{\mathrm{GT}})^2} |\Delta T_z|, \tag{23}$$

where $\Delta T_z = T_z - T_z^{\mathrm{GT}}$. As $T_z^{\mathrm{GT}}$ increases, the effective weight decays most slowly for $\mathcal{L}_{\mathrm{w}}$ and most aggressively for $\mathcal{L}_{\mathrm{inv}}$. In practice, inverse-depth loss may lead to unstable optimization due to excessively large gradients for nearby instances, especially in joint optimization with pose and shape. In contrast, the weighted loss preserves the geometric preference for nearby subjects while providing more stable gradient magnitudes for training.

## B.2. On the Use of Two Depth Losses

In our framework, depth supervision is applied at two different stages, namely the HMR branch and the depth branch. Although both losses involve the per-person pelvis depth $T_z$, they serve distinct purposes and operate under different optimization contexts.

**Depth loss in the HMR branch.** In the HMR branch, the predicted depth $T_z$ is tightly coupled with pose, shape, and camera parameters, and directly affects 2D reprojection through perspective projection. As shown in Eq. 13, we supervise this depth using a focal-length-normalized depth loss formulation. This design accounts for variations in camera intrinsics and enforces a scale-consistent depth representation across samples. Without normalization, the same absolute depth error would induce different reprojection sensitivities under different focal lengths, leading to unbalanced gradients during joint optimization. Normalizing depth by the focal length mitigates this issue and stabilizes training when depth is optimized together with pose and shape parameters.

**Depth loss in the depth branch.** The depth branch is designed to predict per-person pelvis depth directly from full-image features. Here, the depth prediction is optimized independently from mesh regression and camera estimation. Therefore, we apply a direct metric depth supervision without focal-length normalization (Eq. 7), which encourages accurate depth regression in physical units. This branch focuses on learning reliable per-instance depth cues and providing robust depth priors for downstream mesh refinement, rather than ensuring reprojection consistency.

## C. Limitations

The current study focuses on body-only mesh estimation, which can be extended to whole-body recovery in future work. In addition, we train the model using the BED-LAM (1fps) dataset (Black et al., 2023) as a balanced choice between training cost and performance. With sufficient computational resources, using the full BEDLAM (6fps) dataset (Black et al., 2023) and incorporating more real-world multi-person datasets could further improve performance.

## D. Additional Visualization

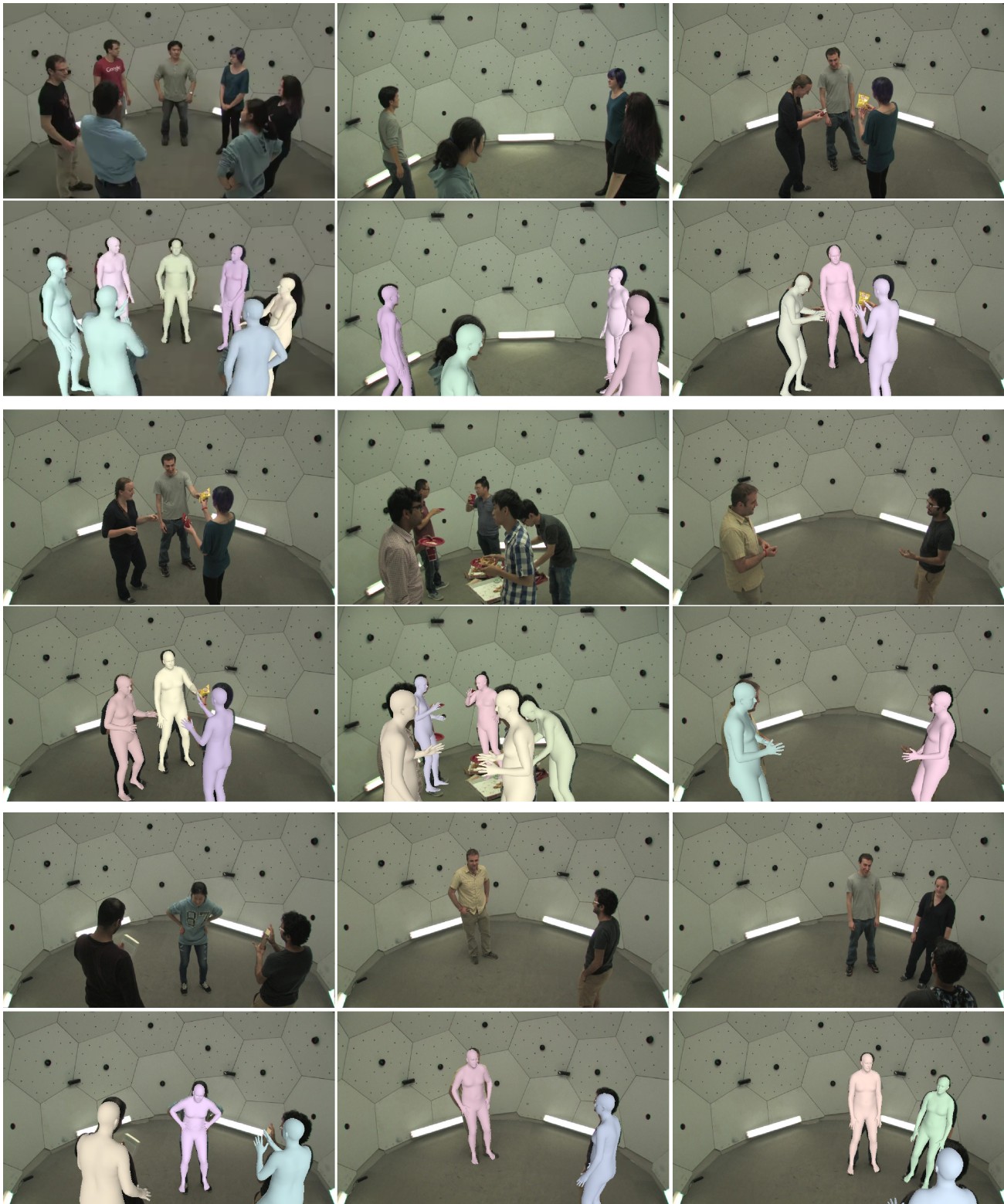

*Figure 5.* Visualization on CMU Panoptic dataset.

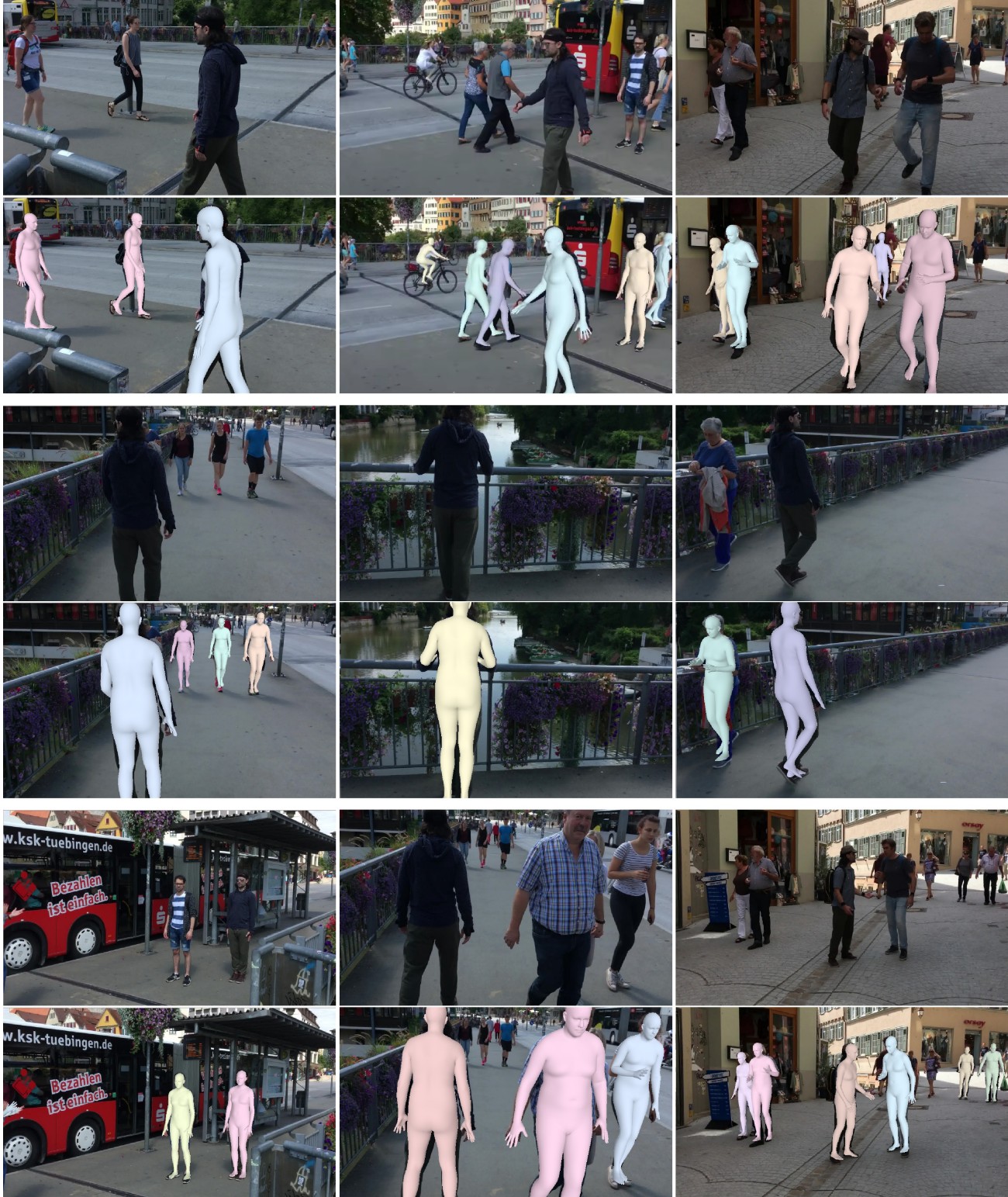

*Figure 6.* Visualization on 3DPW dataset.

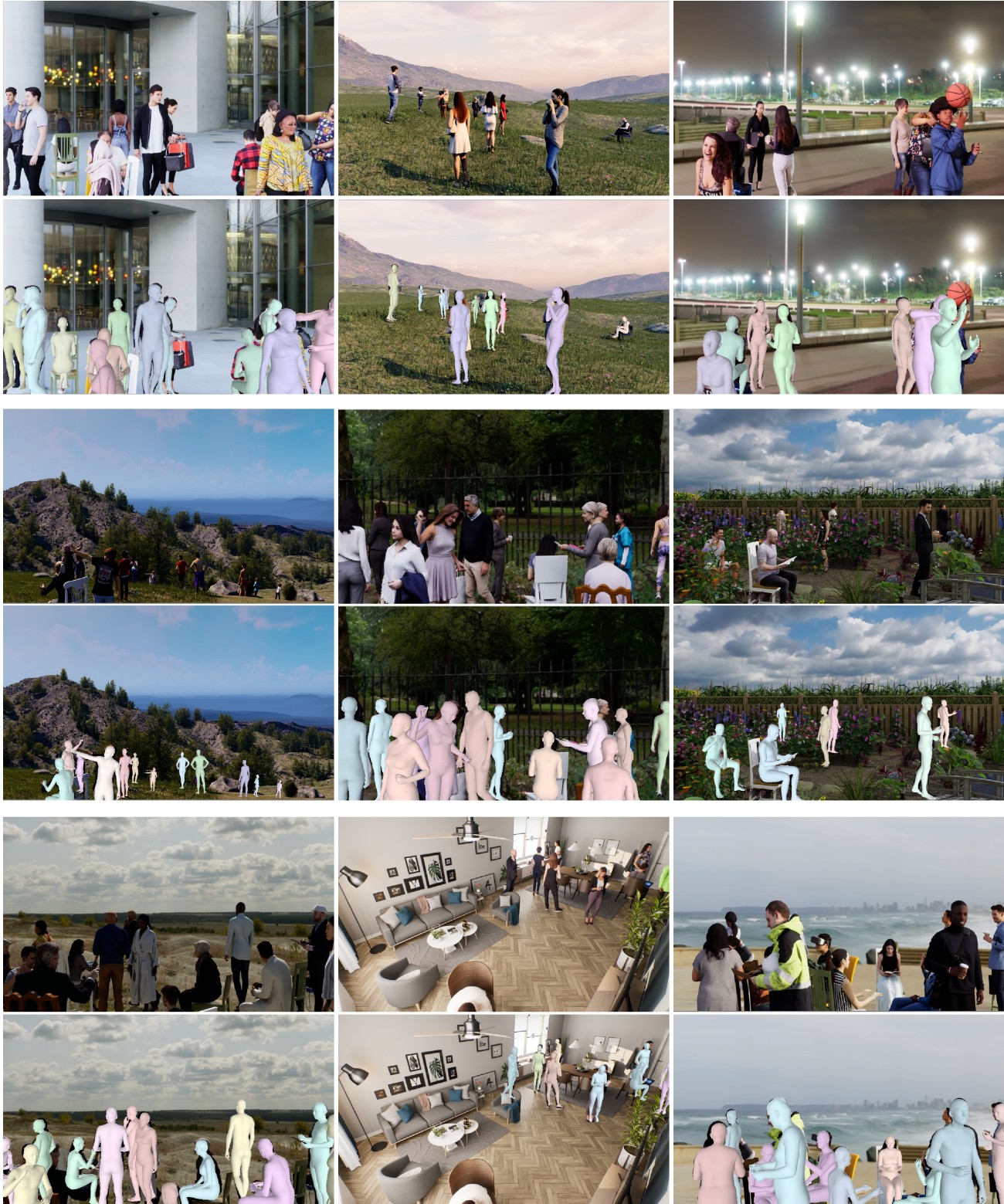

*Figure 7.* Visualization on AGORA dataset.

