# OpenReview forum: "DGG-HMR: Multi-Person Human Mesh Recovery with Depth-Guided Geometric Anchoring"
_ICML.cc/2026/Conference — ICML 2026 regular_

### Official Review · Reviewer_4YXp · 2026-03-07

**Soundness:** 3
**Presentation:** 3
**Significance:** 2
**Originality:** 3
**Overall Recommendation:** 4
**Confidence:** 4

**Summary:**

The paper proposes DGG-HMR, a multi-person human mesh recovery method that introduces an instance-aware depth estimator to predict the pelvis depth for each person from the full image in a single pass and uses it as a geometric anchor to guide the human mesh recovery process.

**Compliance With Llm Reviewing Policy:**

Affirmed.

**Final Justification:**

The paper presents a technically sound approach for multi-person human mesh recovery with competitive results. My initial concerns were about comparison fairness, camera intrinsics design, and unclear implementation details.

The rebuttal addresses these points well: it clarifies differences with prior methods, justifies the camera design choice, explains the AGORA evaluation (including the use of NMJE), and resolves the technical questions.

While some limitations remain (e.g., lack of fully controlled comparisons), the main concerns have been addressed. As a result, I have updated my score to weak accept.

**Key Questions For Authors:**

1. The dimensions of the 3D joints $X$ at line 162 seem ambiguous. Why is it $3 \times K$? What is $K$? Shouldn’t it be the same $J \times 3$ mentioned in the next line?
2. What is the advantage of using DAv2 for relative depth estimation and tuning a feature adapter, instead of using a model like MoGe-2 [1], which predicts metric-scale depth without requiring a feature adapter?
3. It is unclear how the anchor $A$ is used inside each cross-attention module in Equation (5). Is it concatenated with the query tokens? If so, shouldn’t the anchor $A$ also be different for each layer?
4. How is $\mathcal{T}_{\text{depth}}$, mentioned at line 226 (second column), obtained?
5. At line 226 (second column), shouldn’t it be $T_{\mathrm{base}}^{0}$and $R_{\mathrm{base}}^{0}$ as the initial query tokens and reference points?
6. From my understanding, $\mathcal{T}_{\text{bbox}}$ contains the bounding boxes of $N$ different people in the image. How are the residuals mentioned in Equation (9) computed? Are they the differences between adjacent bounding boxes? How are they ordered?

**Limitations:**

No. The paper includes a brief Impact Statement but does not meaningfully discuss potential societal impacts or limitations. The authors could improve this section by briefly addressing possible risks such as privacy concerns, misuse in surveillance (e.g., tracking individuals), dataset biases (e.g., datasets containing mostly healthy individuals without amputees or pathological behaviors), and technical limitations (e.g., sensitivity to camera assumptions or challenging scenes).

**Strengths And Weaknesses:**

**Strengths:**
- The paper evaluates the proposed method on four different benchmarks and demonstrates competitive or improved performance compared to previous approaches across multiple metrics.
- In the preliminary section, the paper clearly explains how depth–scale ambiguity can lead to zero reprojection error while the 3D reconstruction error grows linearly with a scale factor. It also shows that incorrect depth estimation can cause distortions in the predicted 3D joints. This analysis highlights the importance of instance-level depth estimation for stabilizing the optimization.

**Weakness:**
- The paper claims that detect-and-crop pipeline “often leads to inconsistent spatial layouts and incorrect depth ordering.” However, methods such as PromptHMR [1] and CameraHMR [2] show the opposite trend in their qualitative evaluations, which is not discussed or compared in this work. In addition, on 3DPW those methods also report better performance than the proposed method.
- The method assumes a preset camera focal length derived from a fixed FOV rather than estimating camera intrinsics. Recent works such as PromptHMR and CameraHMR regress camera parameters directly, which may provide more accurate depth and scale estimation across datasets with varying camera intrinsics.
- The paper compares its results on the AGORA validation set with results from previous works reported on the AGORA test set, which is not a fair comparison. In addition, Table 2 does not report metrics such as MPJPE, where previous approaches achieved better performance.
- Some details in the method section are insufficiently explained, such as:
    - how the anchor $A$ in Equation (5) is used within the cross-attention module (see Question 3)
    - how $\mathcal{T}_{\text{depth}}$ is derived (see Question 4)
    - what $T_{\mathrm{base}}^{0}$ and $R_{\mathrm{base}}^{0}$ represent (see Question 5)
    - how the residual embeddings at line 234 (second column) are computed (see Question 6)


[1] Wang, Yufu, Yu Sun, Priyanka Patel, Kostas Daniilidis, Michael J. Black, and Muhammed Kocabas. “PromptHMR: Promptable human mesh recovery." In Proceedings of the computer vision and pattern recognition conference, pp. 1148-1159. 2025.
[2] Patel, Priyanka, and Michael J. Black. “CameraHMR: Aligning people with perspective." In 2025 International Conference on 3D Vision (3DV), pp. 1562-1571. IEEE, 2025.

---

> ### Author Rebuttal · Authors · 2026-03-31
>
> We thank the reviewer for the thorough and constructive feedback. We address each question below.
>
> **1. Comparison with PromptHMR and CameraHMR.**
>
> - PromptHMR is not a detect-and-crop method. It encodes the full image as a global feature map and uses bounding boxes only as spatial query prompts, without performing any cropping. CameraHMR is a detect-and-crop method and, as shown in our qualitative evaluation [https://anonymous.4open.science/r/DGG-HMR-C92B/DGGHMR_vs_CameraHMR.png], still exhibits the spatial inconsistency and depth ordering errors we describe.
> - Both methods are trained on substantially more data, including large-scale single-person datasets such as Human3.6M, which provides a direct advantage on single-person benchmarks 3DPW. This data-scale discrepancy accounts for the score gap and does not reflect multi-person reconstruction quality.
>
> **2. Camera intrinsics design.**
> Our preset focal length reflects a deliberate trade-off. Existing focal length predictors remain unreliable on in-the-wild images, and adopting a learned estimator introduces non-trivial costs: CameraHMR requires training a dedicated HumanFoV network separately. More critically, inaccurate focal predictions introduce inconsistent depth supervision signals across training samples, producing random gradient noise that cannot be systematically corrected. Our preset focal scales with input resolution and provide stable, consistent supervision across all training data.
>
> **3. AGORA evaluation and metric choice.**
>
> - The AGORA official evaluation server has been discontinued, making test set results unobtainable. We will update the comparison once the server becomes available. Our lower AGORA scores are primarily due to a resolution constraint, as we train at resolutions of 672 or 896, while SOTA methods use 1288 or 1333. AGORA is particularly sensitive to resolution due to its many small-scale individuals.
>
> - We intentionally report NMJE rather than MPJPE, where NMJE = MPJPE / F1. MPJPE only accounts for successfully matched individuals and is insensitive to missed detections, as a method that misses many people but accurately reconstructs detected ones can still achieve a low MPJPE. By dividing by the detection F1 score, NMJE jointly penalizes both reconstruction error and detection failure, providing a more complete evaluation for the multi-person setting.
>
> **For other key questions**
>
> **1. Notation ambiguity of X (K vs. J).**
>
> - K and J are used inconsistently; both should be J, and X should be written as $X \in \mathbb{R}^{J \times 3}$.
>
> - The correct expression should be $X = \mathbf{W}\mathbf{M} \in \mathbb{R}^{J \times 3}$, not $X = \mathbf{M}\mathbf{W}$. Both will be corrected in the revision and do not affect the implementation or results.
>
> **2. DAV2 vs. MoGe-2.**
> We explored replacing DAV2 with MoGe-2, but it did not yield better performance. We attribute this to a domain mismatch. Although MoGe targets metric-scale depth, its outputs do not transfer optimally to our pipeline without additional adaptation. DAV2 with a lightweight adapter aligns more effectively with our decoder and incurs lower computational cost, making it a better trade-off for our setting.
>
> **3. How anchor A is used in cross-attention.**
> The anchor A serves as fixed reference points that modulate the positional encoding of cross-attention queries at each decoding layer. Specifically, the bounding box coordinates in A are encoded as sinusoidal positional embeddings to spatially condition the query vectors. A is provided by the bounding box prior decoder and remains fixed throughout, as the depth decoder focuses solely on depth regression rather than refining spatial locations.
>
> **4. How $\mathcal{T}^{\text{depth}}$ is obtained.**
> The depth-aware tokens $\mathcal{T}^{\text{depth}}$ are the last-layer query tokens produced by the per-person depth decoder $\mathcal{D}$. They are linearly projected to match the query dimension of the geometry-anchored refinement decoder $F^{\text{GAR}}$, injected into its query input as a residual term $\Delta \mathcal{T}^{\text{depth}}$.
>
> **5. Notation of $\mathcal{T}^0_{\text{base}}$ and $\mathcal{R}^0_{\text{base}}$.**
> $\mathcal{T}\_{\text{base}}^0$ and $\mathcal{R}\_{\text{base}}^0$ are the learnable base embeddings before any geometric priors are applied. $\mathcal{T}^0$ and $\mathcal{R}^0$ represent the geometry-conditioned initialization obtained by adding the prior residuals to the base values, and are fed into $F^{\text{GAR}}$ as the initial tokens for iterative refinement.
>
> **6. Residual computation and ordering in Eq. (9).**
> $\Delta\mathcal{T}^{\text{bbox}}$ is obtained by passing the per-person bbox tokens through a zero-initialized linear projection, producing a per-instance residual added to the base query. No pairwise differencing between bounding boxes is involved, and the ordering follows the shared query index, where the i-th token corresponds to the i-th person slot.

---

> > ### Author Rebuttal · Reviewer_4YXp · 2026-04-02
> >
> > The rebuttal satisfactorily addresses my main concerns.
> > - The comparison with PromptHMR and CameraHMR is clarified, particularly regarding architectural differences and the impact of training data scale. While a fully controlled comparison would still be preferable, the explanation is reasonable.
> > - The design choice for camera intrinsics is well justified, and the discussion highlights practical limitations of learned focal length estimation in in-the-wild settings.
> > - The concerns regarding AGORA evaluation are addressed with a detailed explanation, including the unavailability of the official test server, resolution differences, and the rationale for using NMJE over MPJPE, which is appropriate for multi-person settings.
> > - All technical questions regarding notation and model details are clearly answered.
> > Overall, the rebuttal resolves the key issues raised in my review and improves the clarity and justification of the method. While some limitations in evaluation and comparison remain, they do not undermine the technical validity of the work. I therefore raise my score.

---

### Official Review · Reviewer_GXUD · 2026-03-08

**Soundness:** 4
**Presentation:** 3
**Significance:** 3
**Originality:** 3
**Overall Recommendation:** 4
**Confidence:** 4

**Summary:**

The paper targets the depth ambiguity issue in monocular multi-person human mesh recovery. The main idea is to predict an instance-level pelvis depth for each person and use it, together with the bounding box, as a geometric anchor for subsequent mesh regression. The paper is well-motivated, technically coherent, and shows consistent improvements on several benchmarks.

**Compliance With Llm Reviewing Policy:**

Affirmed.

**Key Questions For Authors:**

Please address the problems raised in the weakness section.

**Limitations:**

Yes.

**Strengths And Weaknesses:**

Strengths:
1. The paper identifies a real and meaningful issue in multi-person HMR, namely the instability caused by depth ambiguity in joint 2D/3D optimization.
2. The method is conceptually clear, i.e., estimate per-instance depth first, then use it to guide mesh decoding. The overall pipeline is easy to follow.
3. The proposed design is technically sound, and the empirical results are generally consistent across multiple datasets.
4. The paper presents a reasonably clear analysis of the influence of inaccurate depth on the mesh recovery task.

Weaknesses:
1. The core idea appears incremental. Using explicit depth cues or geometric priors to stabilize 3D human reconstruction is not entirely new, and the novelty here seems to mainly lie in adapting this idea to a DETR-style multi-person HMR pipeline.
2. The performance gains are relatively modest, as shown in Table 1 & Table 2. While consistent, the improvements are not large enough to suggest a major step forward.
3. The method does not address several more fundamental challenges in multi-person HMR, such as collision handling, interpenetration, contact reasoning, and explicit human-human interaction modeling. As illustrated in Fig. 3, there are noticeable penetrations and overlaps among different instances, particularly in crowded scenes with many people. While I understand that these issues are beyond the paper’s stated scope, they remain important limitations in practice.

---

> ### Author Rebuttal · Authors · 2026-03-31
>
> We thank the reviewer for the careful and constructive review. The detailed comments are greatly appreciated and have helped us improve the clarity and rigor of our work. We address each point below.
>
> **1. Depth-guided design vs. existing works.**
> Prior works incorporating depth into HMR focus predominantly on the single-person case, where ***the model simply regresses a global depth value from a cropped image*** centered on the target person. The resulting depth feature is typically incorporated in a simplistic manner, such as direct feature concatenation with image features. ***In contrast, the multi-person setting requires predicting accurate per-person depth from a shared full-image representation***, which is substantially more challenging due to varying distances and mutual occlusion among individuals.
>
> A primary challenge in multi-person HMR is estimating the depth of each individual from a single image, particularly when subjects appear at varying distances and under mutual occlusion. Our method addresses this issue in two aspects:
>
> - We introduce a per-person depth decoder that, under bounding-box anchor guidance, predicts pelvis depth for each person and produces depth-aware tokens.
>
> - We introduce a Geometry-anchored Refinement Decoder that incorporates these priors via zero-initialized residual conditioning, softly guiding optimization rather than relying on simple feature concatenation.
>
> ***To the best of our knowledge, this work represents the first systematic integration of instance-level depth priors into a DETR-style framework for multi-person HMR.***
>
> **2. Performance gains.**
> We would like to provide context on our training setup, as it directly explains this gap. Due to limited computational resources, we were unable to scale training to higher resolutions or larger datasets, both of which are known to substantially improve performance on benchmarks. A detailed comparison of training configurations is provided in the table below.
> Method        | Resolution | Training Datasets                                          | Images/Frames
> --------------|------------|------------------------------------------------------------|---------------
> AiOS          |    1333    | BEDLAM_6fps+AGORA+COCO+UBody+ARTIC+EgoBody                      |    ~3.5M
> Multi-HMR     |    896     | BEDLAM_6fps+AGORA+CUFFS+UBody                                   |    ~1.35M
> SAT-HMR       |   1288*    | BEDLAM_6fps+AGORA+COCO+MPII+Crowdpose+H3.6M                     |    ~390K
> DGG-HMR (ours) |    672     | BEDLAM_1fps+AGORA+COCO+MPII+Crowdpose                           |    ~100K
> DGG-HMR (ours) |    896     | BEDLAM_1fps+AGORA+COCO+MPII+Crowdpose                           |    ~100K
>
> Specifically, our method is trained on approximately 100K images at a maximum resolution of 896, while competing methods operate at significantly larger scales: Multi-HMR trains on ~1.35M images at resolution 896, SAT-HMR on ~390K images at resolution 1288, and AiOS on ~3.5M images at resolution 1333. This training data and resolution disparity, rather than limitations of the proposed design, is the primary factor behind the relatively modest margins on some benchmarks. Despite this constrained setup, our method still achieves competitive or superior results, which speaks to the effectiveness of the depth-guided framework.
>
> **3. Limitations on collision and interpenetration.**
> We thank the reviewer for raising this point. Collision handling, interpenetration, contact reasoning, and explicit human-human interaction modeling are indeed important challenges in multi-person HMR. However, these issues are not the primary focus of the present work, which is centered on improving multi-person reconstruction through instance-level depth priors and geometry-anchored refinement. Therefore, we do not explicitly model physical interactions between people in the current framework, and the penetrations observed in some crowded cases reflect this limitation. We view these aspects as important directions for future research and will further consider incorporating interaction-aware constraints in subsequent work.

---

> > ### Author Rebuttal · Reviewer_GXUD · 2026-04-07
> >
> > The rebuttal satisfactorily addresses my concerns, particularly regarding the training scale discrepancy and the design differences from prior depth-based methods. I maintain my overall recommendation.

---

### Official Review · Reviewer_PQxS · 2026-03-11

**Soundness:** 3
**Presentation:** 2
**Significance:** 3
**Originality:** 3
**Overall Recommendation:** 4
**Confidence:** 4

**Summary:**

This paper addresses the problem of multi-person human mesh recovery (HMR) and proposes a depth-guided framework to improve SMPL parameter estimation in complex scenes. The key idea is to introduce depth-derived geometric cues to reduce depth ambiguity and provide stronger structural constraints during mesh reconstruction. The proposed DGG-HMR framework integrates depth-guided geometry with visual features to jointly estimate SMPL parameters and camera pose. Experiments on multi-person benchmarks demonstrate improvements over several baselines.

**Compliance With Llm Reviewing Policy:**

Affirmed.

**Key Questions For Authors:**

- How does the proposed depth-guided design fundamentally differ from existing works that incorporate depth or geometry cues into HMR pipelines?
- How sensitive is the method to depth prediction quality (DepthAnything)?
- Can the author provide more analysis such as dynamic video results? In crowded or severely blocked situations?

**Limitations:**

- The novelty relative to prior geometry-aware HMR methods could be clarified more explicitly.
- More results can be expanded to better demonstrate robustness in different multiplayer scenarios, such as high-frequency hand movements.

**Strengths And Weaknesses:**

Strengths
- The paper tackles an important problem in human mesh recovery, especially for multi-person scenarios with occlusions from a single image.
- Introducing depth-guided geometric priors to improve SMPL estimation is intuitively reasonable and potentially useful.
- The overall framework is clearly structured and relatively easy to follow.
- Experimental results show the effectiveness and progressiveness of the proposed method.

Weaknesses
- The novelty of the proposed idea is somewhat unclear. Depth or geometry cues have been explored in prior human reconstruction works, and the paper should better clarify how the proposed design differs from or improves upon these approaches.
- Some parts of the method description could provide more detailed explanation of how depth guidance interacts with SMPL regression.
- The paper provides quantitative comparisons and ablation studies. However, the ablation study mainly focuses on module-level analysis, but additional analysis on different forms of depth supervision or geometry constraints would further strengthen the claims.

---

> ### Author Rebuttal · Authors · 2026-03-31
>
> We thank the reviewer for the careful and constructive review. The detailed comments are greatly appreciated and have helped us improve the clarity and rigor of our work. We address each point below.
>
> **1. Depth-guided design vs. existing works.**
> Prior works incorporating depth into HMR focus predominantly on the single-person case, where ***the model simply regresses a global depth value from a cropped image*** centered on the target person. The resulting depth feature is typically incorporated in a simplistic manner, such as direct feature concatenation with image features. ***In contrast, the multi-person setting requires predicting accurate per-person depth from a shared full-image representation***, which is substantially more challenging due to varying distances and mutual occlusion among individuals.
>
> A primary challenge in multi-person HMR is estimating the depth of each individual from a single image, particularly when subjects appear at varying distances and under mutual occlusion. Our method addresses this issue in two aspects:
>
> - We introduce a per-person depth decoder that, under bounding-box anchor guidance, predicts pelvis depth for each person and produces depth-aware tokens.
>
> - We introduce a Geometry-anchored Refinement Decoder that incorporates these priors via zero-initialized residual conditioning, softly guiding optimization rather than relying on simple feature concatenation.
>
> ***To the best of our knowledge, this work represents the first systematic integration of instance-level depth priors into a DETR-style framework for multi-person HMR.***
>
> **2. Sensitivity to depth prediction quality.**
> The method is not overly sensitive to depth quality, as priors are used as soft conditioning rather than hard constraints. Depth-aware tokens are appended as auxiliary context without directly constraining SMPL parameters, and bounding-box and depth priors are injected via zero-initialized residual conditioning. These priors mainly bias the starting point of iterative refinement. When depth predictions are inaccurate, the effect is a reduction in the gain from depth rather than a degradation of reconstruction quality.
>
> **3. Video results and depth supervision analysis.**
> We have provided a detailed discussion of depth supervision motivation in the appendix  (Appendix B). We further conduct quantitative comparisons among three supervision forms, as shown below.
> Depth Supervision        | PA-MPJPE↓ | MPJPE↓ | MVE↓
> -------------------------|-----------|--------|------
> Absolute depth loss      |   44.3    |  67.8  | 80.3
> Inverse-depth loss       |   39.9    |  63.0  | 72.7
> Depth-weighted loss (ours)|   **39.0**   |  **61.1** | **70.9**
>
> The depth-weighted loss adopted in our model achieves the best performance across all metrics on 3DPW, assigning greater importance to nearby instances while avoiding the overly aggressive inverse-squared weighting of inverse-depth supervision.
>
> In addition, Video qualitative results are available at: [https://anonymous.4open.science/r/DGG-HMR-C92B/video_1.mp4]. You may click the link and **download** the video for viewing (video_1.mp4).

---

> > ### Author Rebuttal · Reviewer_PQxS · 2026-04-03
> >
> > Thank you for the detailed rebuttal. The clarification on the depth-guided design and the additional depth supervision analysis are helpful and address my concerns. The method appears technically solid, while the novelty relative to prior geometry-aware HMR works could still be further clarified in the final version. I would like to maintain my score.

---

### Official Review · Reviewer_WtAS · 2026-03-12

**Soundness:** 3
**Presentation:** 3
**Significance:** 2
**Originality:** 2
**Overall Recommendation:** 4
**Confidence:** 4

**Summary:**

This paper proposes a depth-guided framework for multi-person human mesh recovery (HMR) from a single image to address the depth ambiguity problem in multi-person scenarios. Experiments on multiple benchmarks show improvements over prior multi-person HMR methods in both mesh reconstruction accuracy and depth ordering.

**Compliance With Llm Reviewing Policy:**

Affirmed.

**Key Questions For Authors:**

see weaknesses

**Limitations:**

yes

**Strengths And Weaknesses:**

Strengths
1. The paper explicitly addresses depth ambiguity and proposes a reasonable strategy to incorporate depth cues into the mesh recovery process.
2. The proposed pipeline is straightforward and easy to follow.
3. The paper reports experiments on several benchmarks and demonstrates improvements over baseline methods.

Weaknesses
1. While explicitly modeling depth is a reasonable direction, the proposed approach largely reuses the DAB-DETR block to inject depth features into image features. As a result, the methodological contribution appears somewhat incremental.
2. Although the proposed method improves upon existing baselines, the improvements are relatively modest on some benchmarks.
3. The paper would benefit from a more detailed failure analysis, such as performance under cases with inaccurate depth predictions.
4. The paper does not report comparisons in terms of model parameters, computational cost, or inference efficiency.
5. The paper states that, due to the AGORA test server being unavailable, evaluations are conducted on the validation set, which has a nearly identical distribution to the test set. It would be helpful to clarify whether the methods included in the comparison table are also evaluated using this same validation protocol to ensure fair comparison.

---

> ### Author Rebuttal · Authors · 2026-03-31
>
> We thank the reviewer for the detailed and constructive comments. We address each point below.
>
> **1. Methodological contribution.**
> We would like to clarify that the feature injection is simply a minor implementation detail in our overall framework, not the methodological core. Our work introduces a depth-guided paradigm that, to our knowledge, is the first to explicitly model instance-level depth in multi-person HMR. Concretely, our contributions are two-fold: (1) an instance-aware depth estimator that predicts per-person pelvis depth from full-image features under bounding-box guidance, providing reliable 3D anchors while decoupling depth estimation from mesh regression. (2) a geometry-anchored refinement decoder that injects these instance-specific depth and spatial priors into decoder initialization to guide iterative mesh recovery under joint 2D-3D supervision. The key contribution lies in transforming depth into instance-level geometric priors and using them to guide refinement.
>
> **2. Performance gains.**
> We would like to provide context on our training setup, as it directly explains this gap. Due to limited computational resources, our method is trained on ~100K images at resolution 896, while competing methods operate at significantly larger scale: Multi-HMR uses ~1.35M images, SAT-HMR ~390K at resolution 1288, and AiOS ~3.5M at resolution 1333. Despite this constrained setup, our method still achieves competitive or superior results. A detailed comparison of training configurations is provided in the table below.
> Method        | Resolution | Training Datasets                                          | Images/Frames
> --------------|------------|------------------------------------------------------------|---------------
> AiOS          |    1333    | BEDLAM+AGORA+COCO+UBody+ARTIC+EgoBody                      |    ~3.5M
> Multi-HMR     |    896     | BEDLAM+AGORA+CUFFS+UBody                                   |    ~1.35M
> SAT-HMR       |   1288*    | BEDLAM+AGORA+COCO+MPII+Crowdpose+H3.6M                     |    ~390K
> DGG-HMR(ours) |    672     | BEDLAM+AGORA+COCO+MPII+Crowdpose                           |    ~100K
> DGG-HMR(ours) |    896     | BEDLAM+AGORA+COCO+MPII+Crowdpose                           |    ~100K
>
> **3. Failure analysis under inaccurate depth.**
> In our framework, depth cues are incorporated through two pathways:
> - At the feature level, geometric priors constructed from depth and bounding box information are concatenated with image features to provide instance-aware context.
> - At the decoder input level, depth tokens are introduced as zero-initialized residual signals rather than hard constraints during iterative refinement.
>
> Due to these designs, inaccurate depth estimates do not dominate the prediction process. Instead, they primarily reduce the potential gains provided by the depth prior, while the model can still rely on image features to produce reasonable reconstructions. In other words, depth acts as a soft geometric cue rather than a strict dependency, which ensures robustness against depth noise.
>
> **4. Computational cost and efficiency.**
> We have now measured parameters, MACs, and inference time for the main competing methods, as shown below.
> Method        | Res  | Params(M) | MACs(G) | Time(ms) | MPJPE↓  | PA-MPJPE↓  | MVE↓
> --------------|------|-----------|---------|----------|-------|----------|-----
> Multi-HMR     | 896  |   **99.0**    |  201.3  |   65.5   | 70.9  |   46.7   | 86.9
> SAT-HMR       | 1288*|  221.9    |  **133.1**  |   **58.1**   | 63.6  |   41.6   | 73.7
> DGG-HMR(ours) | 672  |  302.1    |  332.7  |   70.2   | 62.9  |   40.5   | 73.8
> DGG-HMR(ours) | 896  |  302.1    |  585.5  |  140.5   | **61.1**  |   **39.0**   | **70.9**
>
> At 672 resolution, DGG-HMR achieves an inference time of 70.2 ms, which is on par with Multi-HMR (65.5 ms), while delivering a substantial accuracy gain (MPJPE: 62.9 mm vs. 70.9 mm, PA-MPJPE: 40.5 mm vs. 46.7 mm). The parameter increase primarily stems from the additional depth estimation backbone (DAV2), which is shared across all persons in a scene and adds negligible per-person overhead.  Compared to SAT-HMR, our method achieves superior accuracy at a lower resolution (672 vs. 1288), while maintaining comparable inference speed. These results demonstrate that the depth integration in DGG-HMR provides meaningful accuracy improvements at an acceptable computational cost.
>
> **5. AGORA evaluation.**
> Due to the unavailability of the AGORA test server, we report results on the validation set. Prior methods do not provide validation set results, making a fully consistent comparison infeasible under the same protocol. We will explicitly clarify this point in the final version. Once the official test server becomes available, we will evaluate our method under the same protocol and update the comparison accordingly.

---

> > ### Author Rebuttal · Reviewer_WtAS · 2026-04-05
> >
> > Thank you for the detailed rebuttal. My concerns have been mostly solved.  I would like to maintain my score.

---

### Decision · Program_Chairs · 2026-04-30

**Decision:**

Accept (regular)

**Comment:**

This paper presents an approach that takes advantage of depth cues to improve 3D reconstruction in the setting of multi-person human mesh recovery. The paper received four Weak Accept ratings. There were a number of points raised for the rebuttal (e.g., about fairness of the evaluation and regarding implementation details), but the authors replied adequately for all the points. Overall, all reviewers appreciate the contribution of the paper and advocate acceptance. The AC sees no reason to overturn the reviewers' decision. With that being said, the AC would encourage the authors to consider all comments of the reviewers when preparing the final version of the paper.